# The anatomy of unfolding of Yfh1 is revealed by site-specific fold stability analysis measured by 2D NMR spectroscopy

Rita Puglisi [1], Gogulan Karunanithy[2], D. Flemming Hansen[2], Annalisa Pastore [1,3]✉ & Piero Andrea Temussi [1]✉

Most techniques allow detection of protein unfolding either by following the behaviour of single reporters or as an averaged all-or-none process. We recently added 2D NMR spectroscopy to the well-established techniques able to obtain information on the process of unfolding using resonances of residues in the hydrophobic core of a protein. Here, we questioned whether an analysis of the individual stability curves from each resonance could provide additional site-specific information. We used the Yfh1 protein that has the unique feature to undergo both cold and heat denaturation at temperatures above water freezing at low ionic strength. We show that stability curves inconsistent with the average NMR curve from hydrophobic core residues mainly comprise exposed outliers that do nevertheless provide precious information. By monitoring both cold and heat denaturation of individual residues we gain knowledge on the process of cold denaturation and convincingly demonstrate that the two unfolding processes are intrinsically different.

[1] UK-DRI at King's College London, The Wohl Institute, London, UK. [2] Department of Structural Biology, Division of Biosciences, University College London, London, UK. [3] European Synchrotron Radiation Facility, Grenoble, France. ✉email: annalisa.pastore@crick.ac.uk; temussi@unina.it

We are all accustomed to the concept that proteins unfold when the temperature is increased. Less well known is that all proteins unfold in principle also at low temperatures as demonstrated by P. Privalov on purely thermodynamics grounds[1]. According to this theory, the driving force of heat denaturation is the increase of conformational entropy with temperature. This automatically involves the hydrophobic core and disfavours less ordered parts of the architecture. On the contrary, while the mechanism of cold denaturation is still debated, the current hypothesis is that this transition occurs when entropy decreases. In this case, the driving force of unfolding would be driven by the sudden solvation of the hydrophobic residues of the core[1].

The reason why cold denaturation is much less understood than heat transition is that most proteins undergo cold denaturation at temperatures below the water freezing point. This is unfortunate because observation of both unfolding temperatures is in principle very valuable as it allows the calculation of reliable stability curves of the protein and of the whole set of thermodynamic parameters.

We have identified a protein, Yfh1, that, as a full-length natural protein, undergoes cold and heat denaturation at detectable temperatures when in the absence of salt[2]. We have extensively exploited these properties to gain new insights both on the denatured states of Yfh1[3] and on the factors that may influence its stability[4]. The value of Yfh1 as a tool to investigate the unfolding process is evidenced not only by our subsequent work[5–8] but also by papers from other laboratories[9–12].

In our studies, we noticed that most techniques employed to monitor protein stability are however not regiospecific, as they yield a global result, i.e. an estimate of the stability of the whole protein architecture, observable through the global evolution of secondary structure elements upon an environmental insult. This is because we postulate an all-or-none cooperative process in which the protein collapses altogether from a folded to an unfolded state. When monitoring the unfolding of a protein by CD spectroscopy, for instance, we observe intensity changes related to the disruption of alpha helices and/or beta-sheets under the influence of physical or chemical agents.

It would instead be interesting to gauge the response of selected regions of the protein at the single residue level to gain new insights into the mechanisms of the unfolding of selected parts of the protein structure. A technique ideally suited for this purpose is 2D $^{15}$N HSQC spectroscopy since it provides a direct fingerprint of the protein through mapping each of the amide protons. Volume variations of the NMR resonances may reflect changes affecting single atoms of each residue and indirectly report on how they are individually affected by the unfolding process. We recently showed, using Yfh1 as a suitable model, that it is possible to use 2D NMR to measure protein stability and get thermodynamic parameters comparable to those obtained by CD[13]. We showed that this is possible provided that the residues chosen are those buried in the hydrophobic core, thus experiencing the unfolding process directly. To reliably select these residues, we introduced a parameter RAD which was defined as the combination of the depth of an amide group from the protein surface and the relative accessibility at the atom level[13]. We demonstrated that, by excluding most of the exposed residues (RAD values for the amide nitrogens ≥0.5) and averaging over resonances from residues with RAD values lower than 0.1, we can obtain thermodynamics parameters indistinguishable, within experimental error, from those obtained by CD or 1D NMR[13].

Using the approach previously developed[13], we systematically analysed in the current work the heat and cold denaturation of Yfh1 at residue detail but we reversed the perspective and wondered what information, if any, would be carried by residues far from the hydrophobic core and how they reflect the process of unfolding. This subject has increasingly attracted attention: as put in the words of a recent study by Grassein et al.:[14] "For most of the proteins, this global heat-induced denaturation curve can be formally described by a simple two-state (folded/unfolded) statistical model. Agreement with a two-state model does not imply, however, that the macromolecule does not unfold through a number of intermediate states…. Hence, the global denaturation curve hides the heterogeneity of protein unfolding. …Local nativeness is not uniquely defined and is probe dependent." Understanding how individual residues report on protein unfolding is also relevant in view of an increasing number of studies on protein stability based on the intensity variations of the resonance of a single residue upon unfolding[15–17]. The excellent agreement between NMR and CD thermodynamic parameters using 2D NMR[13] put us in the position to examine the output of single residues critically and follow the process of unfolding at an atomic level.

Using once again Yfh1, we show here that it is possible to sort out which individual single residues yield stability curves consistent with the global unfolding process and that we can obtain valuable information on the process of unfolding from residues that diverge from the average behaviour: whereas some of the residues signal a single folding/unfolding event, we find that others report on more complex thermodynamic events. Our data directly demonstrate that the cold and heat denaturation processes have distinctly different mechanisms and provide site-specific information on solvent interactions supporting Privalov's interpretation of cold denaturation[1]. Our results also clearly demonstrate the considerable advantages of NMR over other approaches, such as in CD or fluorescence, that probe only bulk transitions or individual residues.

## Results

**Data collection and preliminary considerations**. To study the unfolding of Yfh1, we collected $^{15}$N HSQC spectra of Yfh1 at different temperatures and extracted the volumes of individual residues as a function of temperature (Supplementary Fig. S1). This could be confidently done for 68 (out of the expected 109) well-resolved resonances. The behaviour of $^{15}$N HSQC spectra of Yfh1 as a function of temperature was not uniform: some peaks could be observed nearly at all temperatures in the range 278–313 K, others disappeared at temperatures intermediate between room temperature and the two unfolding temperatures, i.e. lower than 313 K or higher than 278 K (Supplementary Figs. S2, S3). This behaviour can of course be ascribed to the exchange regime (intermediate) between folded and unfolded conformations of these residues and told us that they are not an integral part of the architecture of the folded form. The possibility that the intensity changes in the HSQCs at low temperature could be solely due to exchange broadening and not to unfolding can however be excluded by the practically perfect agreement between the curves obtained by CD and by NMR (both 1D[6] and 2D[13]). Cold denaturation of Yfh1 has also been independently confirmed by five independent techniques[14–16].

**Extraction of the thermodynamics parameters**. We could then extract the thermodynamic parameters of the unfolding process for the selected resonance assuming that some conditions are met[1,4]. We first assumed that unfolding transitions are, at a first approximation, two-state processes from folded (F) to unfolded (U) states. We then postulated that the difference in the heat capacity of the two forms ($\Delta C_p$) does not depend on temperature.

This assumption is considered reasonable when the heat capacities of the native and denatured states change in parallel with temperature variations[1]. When these two conditions are reasonably met, the populations of the two states at temperature $T$, $f_F(T)$ and $f_U(T)$, are a function of the Gibbs free energy of unfolding, $\Delta G^o(T)$ (see Methods and Martin et al.)[4]. The plot of the free energy of unfolding as a function of temperature provides what is called the stability curve of a protein[18]. From this equation the main thermodynamic parameters, i.e. heat melting temperature ($T_m$), enthalpy difference at the melting point ($\Delta H_m$) and the heat capacity difference at constant pressure ($\Delta C_p$), can be determined using a non-linear fit (damped least-squares method, also known as the Levenberg–Marquardt algorithm)[19,20]. Other parameters, e.g. the low-temperature unfolding ($T_c$), can be read from the stability curve. When the original assumptions are significantly wrong, fitting results in unrealistic numbers. In our case, the volumes were transformed into relative populations of folded Yfh1 assuming that, as measured by CD and confirmed in other studies on Yfh1[2,4,7,8], unfolded forms are in equilibrium with, on average, a 70% population of folded Yfh1 at room temperature. The concurrent presence of an equilibrium between folded and unfolded species of Yfh1 at low ionic strength was proven by the coexistence of minor extra peaks which disappear as soon as physiologic concentrations of salt are added[21].

**Identification of residues consistent with or outliers from the global behaviour.** We correlated each amide resonance to the corresponding value of RAD, the parameter introduced in Puglisi et al.[13], to pinpoint residues close to the hydrophobic core (Table 1). Of the 68 residues selected, 39 had RAD <0.5, 37 with RAD <0.4, 33 with RAD <0.3, 24 with RAD <0.2 and 11 RAD <0.1 (Table 1). The residues with RAD <0.1 (henceforth called RAD_0.1) were used to calculate the average. Comparison of the stability curves of the nonoverlapping amide resonances with this average showed that several residues with quite different RAD values yield stability curves drastically different from the average (Fig. 1a). We next tried to classify the individual stability curves into those that matched well the average RAD_1 curve ('well-behaved') and those that did not ('ill-behaved'). The curves for residues in the hydrophobic core were in good agreement with the average curve (Fig. 1b).

However, we could not find in general a clear-cut criterion to decide when the curves were not consistent with the average. We arbitrarily chose to set a cut-off at values of the unfolding temperatures ($T_m$ and $T_c$) that differed, on average, less than 1.5 K from those corresponding to the average (RAD_0.1). An alternative choice could be using the Ts but this would report only on one temperature whereas we have shown before that the best parameter to describe protein stability is the whole area of the stability curve between $T_c$ and $T_m$[8].

The $\Delta T_m$ and $\Delta T_c$ differences are smaller than the variability that we had observed among different preparations and measurements of the same protein[2,4,7,8,13,22]. The residues selected according to this criterion are E71, E75, D78, L91, D101, L104, M109, T110, Y119, I130, L132, F142, D143, L152, L158, T159, D160 and K168 (Fig. 1c). Most of the amide groups of the well-behaved residues are spread among well-structured secondary elements, but a few are in less ordered regions (Fig. 2a). By the same token, we selected as 'ill-behaved' residues those whose $T_m$ and $T_c$ values differed from the average curve, on average, more than 3 K with respect to the best curve RAD_0.1. Twenty-one residues (V61, Q63, H83, L88, S92, H95, C98, I99, G107, V108, I113, V120, N127, K128, Q129, L136, N146, G147,

**Table 1 Thermodynamic parameters of the detectable residues.**

| | $\Delta H$ (Kcal/mol) | $\Delta S$ (Kcal/mol) | $\Delta C_p$ (Kcal/molK) | $T_m$ (K) | $T_c$ (K) | RAD |
|---|---|---|---|---|---|---|
| 61 Val | 19.9 | 0.067 | 1.58 | 298.4 | 273.9 | 48.20 |
| 63 Gln | 21.1 | 0.072 | 2.24 | 294.0 | 275.6 | 3.64 |
| 64 Glu | 27.7 | 0.093 | 3.30 | 298.0 | 281.5 | 2.52 |
| 65 Val | 24.3 | 0.081 | 2.71 | 299.7 | 282.2 | 0.31 |
| 68 Leu | 21.0 | 0.071 | 3.02 | 296.5 | 282.8 | 0.75 |
| 70 Leu | 28.5 | 0.096 | 3.73 | 298.2 | 283.1 | 2.35 |
| 71 Glu | 29.1 | 0.097 | 3.59 | 299.8 | 283.9 | 7.13 |
| 72 Lys | 33.1 | 0.111 | 3.73 | 299.6 | 282.2 | 6.24 |
| 75 Glu | 24.3 | 0.081 | 2.75 | 300.0 | 282.7 | 2.07 |
| 76 Glu | 17.4 | 0.058 | 1.71 | 300.6 | 280.8 | 0.91 |
| 78 Asp | 29.0 | 0.096 | 3.18 | 301.1 | 283.2 | 0.17 |
| 83 His | 22.8 | 0.076 | 2.21 | 298.4 | 278.2 | 0.21 |
| 86 Asp | 25.4 | 0.084 | 2.62 | 301.0 | 282.0 | 0.27 |
| 87 Ser | 22.8 | 0.076 | 2.42 | 299.1 | 280.6 | 0.34 |
| **88 Leu** | **20.2** | **0.067** | **1.75** | **300.6** | **278.1** | **0.04** |
| 89 Glu | 22.1 | 0.074 | 2.14 | 299.7 | 279.5 | 0.20 |
| 90 Glu | 26.1 | 0.087 | 2.50 | 300.9 | 280.5 | 0.52 |
| 91 Leu | 34.1 | 0.114 | 4.19 | 300.3 | 284.3 | 0.16 |
| 92 Ser | 22.6 | 0.075 | 1.97 | 299.7 | 277.3 | 0.15 |
| 93 Glu | 19.5 | 0.065 | 1.97 | 300.1 | 280.7 | 0.61 |
| 94 Ala | 23.6 | 0.079 | 2.55 | 299.6 | 281.5 | 4.10 |
| 95 His | 18.6 | 0.062 | 1.31 | 300.7 | 273.2 | 0.28 |
| 97 Asp | 23.3 | 0.078 | 2.52 | 299.0 | 280.9 | 0.95 |
| 98 Cys | 22.6 | 0.076 | 2.59 | 297.7 | 280.6 | 0.26 |
| 99 Ile | 17.6 | 0.059 | 1.75 | 297.5 | 277.8 | 0.11 |
| 101 Asp | 22.3 | 0.074 | 2.42 | 300.2 | 282.1 | 1.08 |
| 104 Leu | 29.5 | 0.098 | 3.12 | 300.9 | 282.4 | 0.78 |
| 105 Ser | 23.7 | 0.079 | 2.51 | 299.7 | 281.2 | 1.18 |
| 107 Gly | 19.3 | 0.065 | 2.44 | 296.6 | 281.1 | 3.58 |
| 108 Val | 22.3 | 0.075 | 2.03 | 299.0 | 277.5 | 0.50 |
| 109 Met | 26.6 | 0.089 | 3.25 | 299.4 | 283.3 | 0.63 |
| 110 Thr | 33.2 | 0.111 | 3.79 | 300.2 | 283.0 | 0.23 |
| 112 Glu | 28.5 | 0.094 | 2.88 | 301.9 | 282.5 | 0.41 |
| 113 Ile | 17.4 | 0.059 | 2.11 | 297.6 | 281.3 | 0.12 |
| 115 Ala | 15.4 | 0.052 | 2.77 | 295.8 | 284.9 | 2.48 |
| 116 Phe | 14.8 | 0.049 | 1.73 | 298.0 | 281.3 | 0.62 |
| 117 Gly | 27.6 | 0.093 | 3.48 | 298.2 | 282.6 | 0.98 |
| 119 Tyr | 24.9 | 0.083 | 2.72 | 300.3 | 282.3 | 0.22 |
| 120 Val | 15.7 | 0.053 | 1.68 | 297.3 | 279.0 | 0.33 |
| 127 Asn | 23.0 | 0.077 | 2.46 | 297.0 | 278.7 | 5.81 |
| 128 Lys | 15.5 | 0.052 | 1.35 | 300.4 | 277.9 | 0.66 |
| 129 Gln | 14.3 | 0.048 | 1.80 | 296.8 | 281.2 | 0.20 |
| **130 Ile** | **27.9** | **0.093** | **3.19** | **300.9** | **283.8** | **0.02** |
| **131 Trp** | **21.7** | **0.073** | **2.54** | **298.6** | **281.8** | **0.04** |
| **132 Leu** | **26.4** | **0.088** | **3.34** | **300.7** | **285.1** | **0.02** |
| 133 Ala | 24.7 | 0.082 | 2.52 | 299.9 | 280.7 | 0.19 |
| 134 Ser | 20.3 | 0.068 | 2.07 | 299.7 | 280.5 | 0.13 |
| 136 Ser | 13.2 | 0.044 | 1.27 | 299.3 | 278.9 | 0.25 |
| 140 Asn | 25.4 | 0.085 | 2.80 | 299.4 | 281.6 | 0.17 |
| **142 Phe** | **20.9** | **0.070** | **3.06** | **299.3** | **285.8** | **0.03** |
| 143 Asp | 21.9 | 0.073 | 2.50 | 300.2 | 283.1 | 0.13 |
| 146 Asn | 23.6 | 0.080 | 3.61 | 295.0 | 282.1 | 2.00 |
| 147 Gly | 25.2 | 0.085 | 2.37 | 297.8 | 277.0 | 4.80 |
| 148 Glu | 21.6 | 0.072 | 2.69 | 298.8 | 283.0 | 1.40 |
| **150 Val** | **22.9** | **0.076** | **2.20** | **300.7** | **280.4** | **0.03** |
| **151 Ser** | **22.7** | **0.076** | **2.39** | **299.9** | **281.3** | **0.05** |
| 152 Leu | 32.2 | 0.107 | 3.87 | 300.0 | 283.7 | 0.16 |
| 154 Asn | 21.9 | 0.074 | 2.40 | 295.1 | 277.2 | 1.14 |
| **158 Leu** | **29.1** | **0.096** | **3.11** | **301.9** | **283.6** | **0.03** |
| **159 Thr** | **29.8** | **0.099** | **3.38** | **300.0** | **282.8** | **0.09** |
| 160 Asp | 23.6 | 0.078 | 2.55 | 301.2 | 283.1 | 0.28 |
| **161 Ile** | **15.4** | **0.051** | **1.66** | **300.1** | **281.9** | **0.09** |
| 163 Thr | 15.2 | 0.051 | 1.44 | 300.9 | 280.2 | 0.15 |
| **166 Val** | **27.3** | **0.091** | **2.81** | **300.8** | **281.8** | **0.06** |
| 168 Lys | 17.3 | 0.058 | 1.93 | 299.9 | 282.4 | 0.16 |
| 170 Ile | 22.6 | 0.075 | 2.11 | 301.3 | 280.3 | 0.31 |
| 172 Lys | 28.1 | 0.095 | 3.84 | 294.1 | 279.7 | 1.5 |
| 174 Gln | 20.6 | 0.069 | 2.2 | 297.5 | 279.2 | |
| 131 Trp sc | 27.1 | 0.091 | 3.21 | 299.4 | 282.8 | |

Rows in boldface indicate residues with RAD values less than 0.1 (RAD_0.1) used to calculate the average stability curve.
The average stability curve was obtained by selecting the residues with RAD_0.1 (Puglisi et al., 2020).

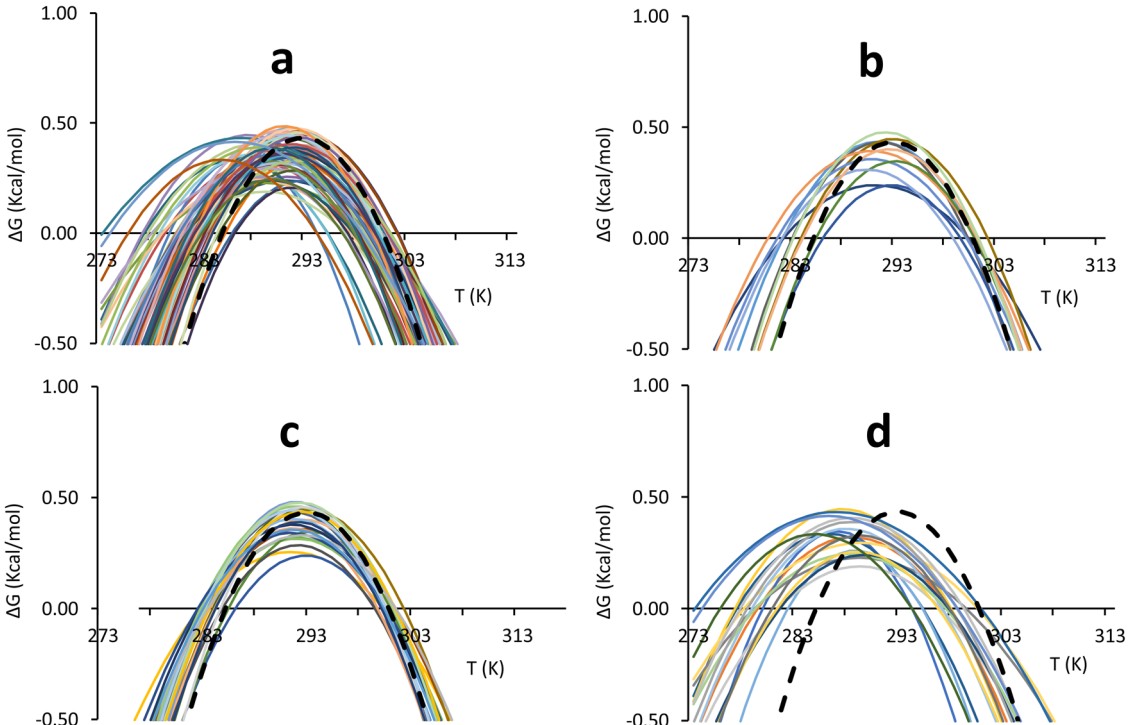

**Fig. 1 Comparison of single residue stability curves with the global RAD_0.1 best curve.** Individual curves are indicated with different colours, the global RAD_0.1 curve is indicated by a dashed black line. **a** Stability curves of all observable isolated residues. **b** Stability curves of residues with a RAD <0.1. **c** Stability curves of single residues for which the difference in the unfolding temperatures with respect to values of the reference curve ($\Delta T_m$ and $\Delta T_c$) is on average below 1.5 °C **d** Stability curves of single residues for which the difference in the unfolding temperatures with respect to values of the average curve ($\Delta T_m$ and $\Delta T_c$) is on average above 3 K. The colour coding is defined in Supplementary Table S1. Note that for simplicity, colour coding was automatically set by the excel software and is not the same in the different panels.

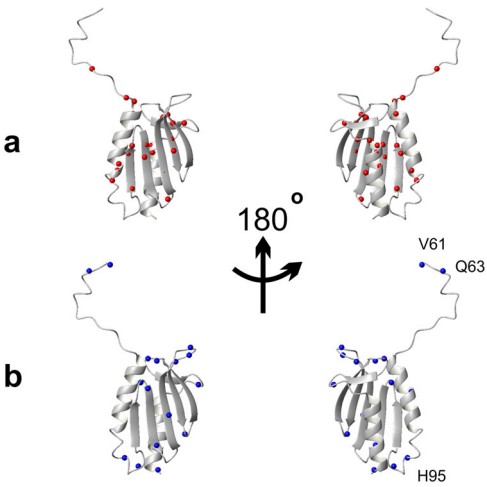

**Fig. 2 Distribution of residues on the structure of Yfh1. We used the reference pdb file 2fql. a** Distribution of the nitrogen atoms of residues for which the difference in the unfolding temperatures with respect to values of the RAD_0.1 curve ($\Delta T_m$ and $\Delta T_c$) is on average below 1.5 K. **b** Distribution of the N atoms of residues for which the difference in the unfolding temperatures with respect to values of the average curve ($\Delta T_m$ and $\Delta T_c$) is on average above 3 K. Indicated explicitly are the three residues whose stability curve is most shifted to lower temperatures with respect to the average RAD_0.1. The structure pairs are rotated by 180 degrees around the y axis.

N154, K172 and Q174) belong to this subset. Except for a few outliers, they are all in less structured regions (Fig. 2b). Amongst these residues, V61, Q63 and H95 which are positioned in flexible regions (either in the N-terminal tail or in a loop), are those with

the largest shift of $T_c$. This behaviour is, however, not a general rule as some of the best-behaved residues reported in Fig. 1c are not in regular secondary structure elements confirming the complexity of the system under study.

The stability curves of the residues that differ from the average (Fig. 1d) have an important peculiarity: most stability curves show a moderate decrease of $T_m$ ($\Delta T_m < 0$) and a large decrease of $T_c$ ($\Delta T_c \ll 0$) from the average. This finding would imply that the corresponding transition temperatures for the heat and cold unfolding point to a decreased stability for heat denaturation but increased stability for cold denaturation.

**Evaluating the contribution of errors.** To make sure that the effect is beyond experimental errors, we reasoned that three phenomena could potentially lead to erroneous populations, $f_F(T)$ and $f_U(T)$, and thus stability curves: (1) the folding exchange dynamics leading to a time-dependent fluctuation of the $^1H$ chemical shift and loss of intensity during the INEPTs of the $^{15}N$-HSQC, (2) differential intrinsic relaxation rates in the folded and unfolded states and (3) exchange of the detected amide protons with the bulk solvent. We thus performed simulations to evaluate how much these phenomena could influence the resulting curves (for a more detailed discussion see the par. "*Simulations to estimate the accuracy of the derived stability curves*" in Supplementary Information). We found that, although the three contributions affect the derived populations, the stability curves that are naïvely calculated from the intensities observed in the NMR spectra as $\widetilde{\Delta G}(T) = -RT \ln\left((1 - I_f)/I_f\right)$, where $I_f$ is the peak intensity of the folded species, recapitulate the general features of the expected stability curve, $\Delta G(T)$. Of particular interest is that the temperature of maximum stability $T_S$ (so-called

because it corresponds to zero entropy of the stability curve), is well reproduced despite the deviations observed for the other parameters (Supplementary Fig. S4).

Our observations are thus beyond experimental error and indicate that the mechanisms of the two unfolding processes, at high and low temperatures, are intrinsically different in agreement with Privalov's theory[1].

**A possible classification of the outliers.** The negative values of $\Delta T_m$ and $\Delta T_c$ observed for some residues (Fig. 1d) imply that also the temperature of maximum stability $T_S$ for these residues is lower than that observed for the best average RAD_0.1. A shift of $T_S$ towards higher temperature values, when studying several cases of thermophilic proteins, was attributed by Razvi & Scholtz[23] to a decrease in the entropy difference in unfolding. Obviously, a *decrease* of $T_m$ or $T_c$ caused by shifting the $T_S$ to lower temperatures is connected to an increase in the entropy difference. This interpretation is based on the classification by Nojima et al.[24] of the main mechanisms of changing the thermal resistance, that is the resistance of heat to cross a material, of a protein. According to the rough classification of Nojima et al.[24], altered thermostability can be achieved thermodynamically in three extreme cases (Fig. 3a). Real situations might of course contain mixtures of the three possibilities.

According to mechanism I, when $\Delta H_S$ (the change in enthalpy measured at $T_S$) increases, the stability curve retains the same shape, but with greater $\Delta G$ values at all temperatures. With mechanism II, a decreased $\Delta C_p$ leads to a broadened stability curve retaining the same maximum, because the curvature of the stability curve is given by $\frac{\partial^2 \Delta G}{\partial T^2} = -\frac{\Delta C_p}{T}$[18]. According to mechanism III, the entire curve can shift towards higher or lower temperatures. It is possible to show[1] that:

$$T_S = T_m \cdot \exp\left[-\frac{\Delta S_m}{\Delta C_p}\right] = T_m \cdot \exp\left[-\frac{\Delta H_m}{T_m \cdot \Delta C_p}\right] \quad (1)$$

Increasing the difference in entropy between the folded and unfolded states ($\Delta S_m$) can shift the values of $T_S$ towards lower temperatures. Most of the curves in Fig. 1d do not correspond to a single mechanism, but to a combination of them (Fig. 3b). Nevertheless, all curves are shifted towards lower values of $T_S$ and larger low-temperature differences correlate well with less ordered regions of the structure. It is thus not surprising to find this behaviour for residues at the N- and C-termini (Q63 and K172) or in connecting loops (G107, N127, N146 and N154) which are bound to be flexible[25]. More surprising is, however, to find amongst these residues also V120 which is right in the middle of the beta-sheet. While we do have not a definite explanation for this observation at the moment, it could indicate a local frustration point in this region.

**Exploring the correlation between stability and secondary structure elements.** We have previously shown that, in addition to the criteria of depth and exposition, an alternative selection of residues over which average populations might be based on elements of regular secondary structure[13]. It is now possible to analyse the behaviour of each secondary structure element. Of the 68 residues selected, 35 were in secondary structure elements (15 in alpha helices, 20 in beta sheets). The largest number of residues of secondary structure traits whose resonance is accessible belongs to the two helices (Fig. 4).

Several resonances have stability curves far from the reference one (dashed black curve of RAD_0.1). These are those of His83 and Leu88 for helix 1 (Fig. 4a). All the others are in fair

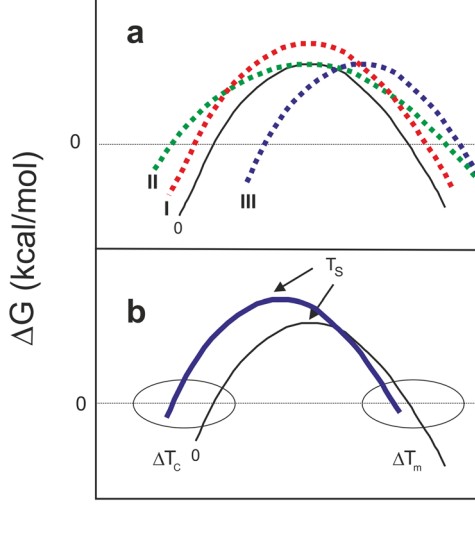

**Fig. 3 Mechanisms that influence stability curves of a protein. a** Dependence of the difference of free energy between unfolded and folded states (**ΔG**) of a hypothetical protein vs temperature (*T*) (curve 0). Mechanism I illustrates the effect of increasing $\Delta H_S$ (curve I). Mechanism II shows the effect of reducing $\Delta C_p$ (curve II). Mechanism III shows the shift of the whole stability curve towards higher temperatures caused by decreasing $\Delta S_m$ (curve III). **b** A combination of the three mechanisms. The solid blue curve, with a prevalent low shift of $T_S$, corresponds qualitatively to the cases of Yfh1 reported in Fig. 1d. The figure was adapted from Nojima et al., 1977.

agreement with the average curve. The best-behaved residue (Asp78) is located at the end of the helix with its amide groups in the buried side of the helix. For helix 2, the worst agreement is found for Thr163 and Ile170, whereas the best agreement is for Leu158, Thr159, Asp160 and Lys168 (Fig. 4b). This implies that residues of helix 2 with a good agreement are distributed over the whole secondary structure element. Some residues of helix 2 have also lower stability curves which indicate a lower $\Delta H$.

The number of residues belonging to beta strands for which it was possible to extract stability curves is more limited (Fig. 5).

The best agreement was found for Leu104 of strand 1, Met109 and Thr110 of strand 2, Tyr119 for strand 3, Ile130 and Leu132 of strand 4 and Phe142 and Asp143 of strand 5.

**The behaviour of tryptophan side chains.** We then looked into the possibility of following the process of unfolding and calculating thermodynamic parameters using the tryptophan side chains. This choice directly parallels studies based on following the process of unfolding by fluorescence using the intrinsic tryptophan fluorescence[26]. Yfh1 has two tryptophans: W131 is fully exposed to the solvent whereas W149 is buried. Both residues are fully conserved throughout the frataxin family and the two side-chain resonances are clearly identifiable (Supplementary Fig. S5a). We calculated the thermodynamic parameters for the side chain indole groups of both residues by the same procedure outlined for main chain NHs, generating first a stability curve. The resonance of W149, which could potentially be more interesting, could not be used for quantitative measurements because the temperature dependence of its volume yields a stability curve very different from the others (Supplementary Fig. S5b) and leads to impossible fitting parameters. This might be explained by the coexistence of folded

and partially unfolded species in equilibrium with each other in solution. As a consequence, the indole of W149 resonates both at 9.25 and 127.00 ppm (folded species) and at ca. 10.05 and 129.20 ppm (split into three closely adjacent peaks, unfolding intermediates) (Supplementary Fig. S5a). As previously proven experimentally, the resonances of the unfolding intermediates disappear upon the addition of salt (Fig. 1, panel A and B in Vilanova et al.[21]). These resonances are also at the same chemical shifts observed for

the tryptophan indole groups at low and high temperatures where however the three signals collapse into one (Supplementary Fig. S1). The complex equilibrium between different species could thus explain the ill-behaviour of the corresponding stability curve of this residue.

The behaviour of the resonance of the exposed W131 side chain is instead fully consistent with that of RAD_0.1 and also with the original curve calculated from 1D NMR data[2] (Table 1). On the whole, these results exemplify well the complexity of the selection choice of the unfolding reporter and advocate in favour of a wholistic analysis of all the available data.

## Discussion

The de facto demonstration that it is possible to reliably measure the thermodynamic parameters of protein unfolding by 2D NMR spectroscopy[13] has opened a new territory to study protein unfolding at atomic resolution using site-specific information. Following protein folding/unfolding looking at specific residues rather than obtaining an average overall picture is not a novelty. Despite some intrinsic limitations, fluorescence has, for instance, been used for decades to probe protein unfolding following the intrinsic tryptophan fluorescence[26,27]. Another elegant, although sadly still underexploited technique able to report local behaviour at the level of specific residues is chemically induced dynamic nuclear polarisation (CIDNP), first introduced to the study of proteins by Robert Kaptein[28]. This technique allows the selective observation of exposed tryptophans, histidines and tyrosines. In protein folding, it was, for instance, used to characterise the unfolded states of lysozyme[29,30] and the molten globule folding intermediate of α-lactalbumin[31,32]. Real-time CIDNP was also used to study the refolding of ribonuclease A[33] and HPr[34]. The only drawback of this technique is that, as in fluorescence, the information is limited to specific aromatic residues.

Another important technique that reports on protein unfolding at the single residue level is stopped-flow methods coupled with NMR[35,36] or mass spectrometry measurements[37] of hydrogen

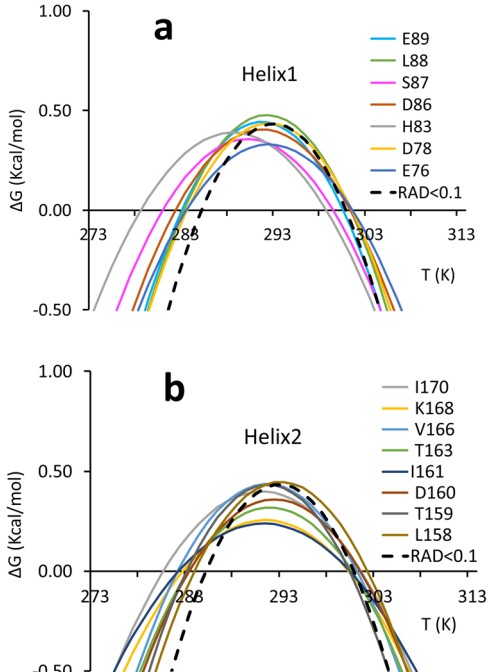

**Fig. 4 Stability curves of residues belonging to secondary structure elements. a** Helix 1. **b** Helix 2. Residues are labelled with a single letter code. The average stability curve is shown as a black dashed line.

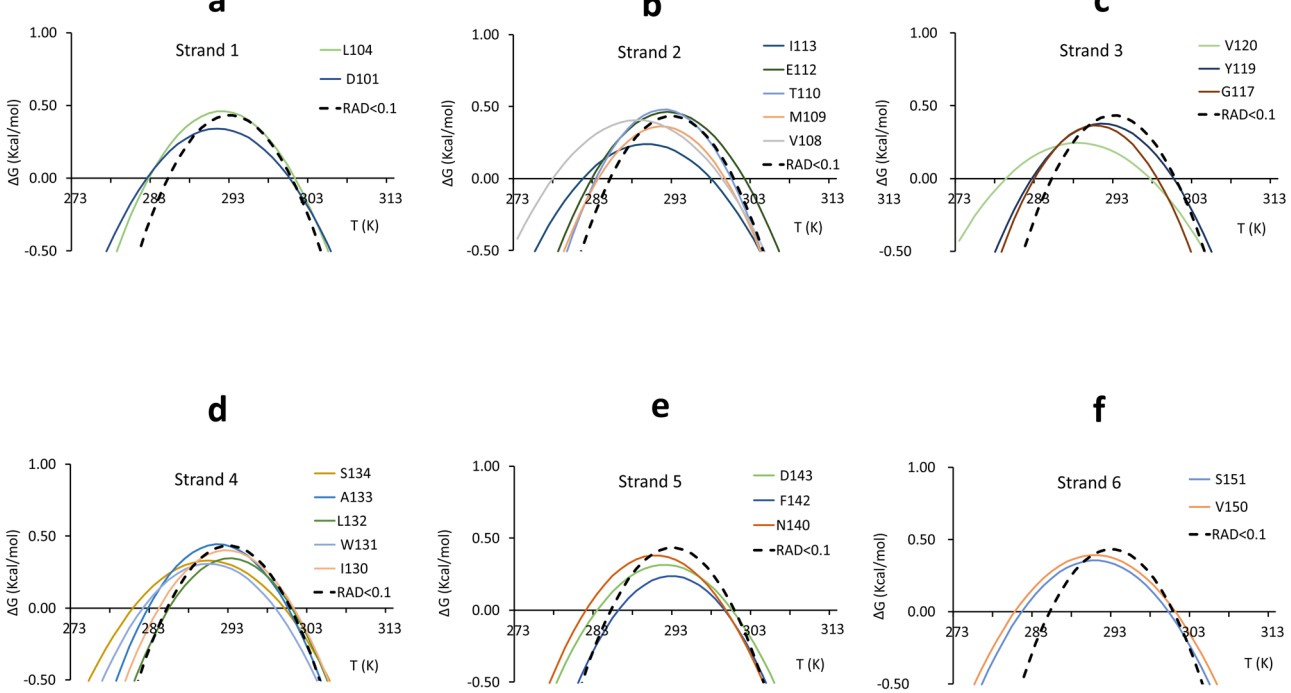

**Fig. 5 Stability curves of residues belonging to secondary structure elements. a** Strand 1. **b** Strand 2. **c** Strand 3. **d** Strand 4. **e** Strand 5. **f** Strand 6. Residues are labelled with a single letter code. The average stability curve is shown as a black dashed line.

exchange. In a classic paper[38], Dobson and co-workers described, for instance, NMR experiments based on competition between hydrogen exchange as observed in COSY spectra and the refolding process. The authors concluded that the two structural domains of lysozyme followed two distinct folding pathways, which significantly differed in the extent of compactness in the early stages of folding. Similar and complementary conclusions could be reached by integrating NMR with mass spectrometry[37]. While these studies retain their solid importance, the possibility of following the resonance intensities also by HSQC spectra may provide a more flexible tool to obtain detailed information on unfolding, as this technique reports on the exchange regime but also, implicitly, on the chemical environment. The use of 2D HSQC had been discouraged by the non-linear relationship between peak intensity (or volume) and populations with temperature as the consequence of relaxation, imperfect pulses, and mismatch of the INEPT delay with specific J-couplings. We have previously suggested an approach to compensate for these effects and demonstrated that the non-linearity does not affect the spectra of Yfh1[13], even though these conclusions might be protein-dependent.

Here, we used the approach developed in our previous work[13] to analyse individual stability curves for most of the residues of Yfh1. Our analysis is highly complementary to the single residue information that may be obtained through HDX by NMR or mass spectrometry[39,40]. A clear advantage of the current approach is the availability of signals of almost all residues and the relative simplicity of the analysis.

We noted that Yfh1 shows a multitude of events on top of the overall folding/unfolding. We observed that the behaviour of the individual stability curves is not distributed uniformly along the sequence. Residues can be clearly divided into two groups, i.e. those consistent with the average behaviour of an all-or-none mechanism of unfolding and those differing, even strongly, from the best average RAD_0.1. This finding alone proved that it is not possible to measure stability using a single residue without a careful evaluation of the role of the specific residue in the protein fold. This conclusion is partially mitigated by our results on the parameters obtained for a tryptophan indole. However, on the whole, also for these side chains, it may be difficult, a priori, to infer which tryptophan is more reliable. We showed that of the two tryptophans present in Yfh1 only the fully exposed W131 is suitable for the analysis. Our results thus demonstrate that unfolding studies based on fluorescent measurements using the intrinsic fluorescence of tryptophan should always be taken with a pinch of salt: in many cases, no independent controls are feasible to evaluate the accuracy of the results. The possibility of using 2D NMR and the introduction of the easily approachable RAD parameter may assist in this choice in future studies.

Analysis of individual secondary structure elements, i.e. helices and strands, showed that there is no clear hierarchy among them, and there is no indication that any of the elements undergo disruption before the others, either at high or low temperature. This implies that, overall, the folding/unfolding of the core of Yfh1 can be described as a single, highly cooperative event, but not all residues could be used for following the transition. It will be interesting in the future to study lysozyme to have an example in which two subdomains unfold independently[38]. In addition to information on regular secondary structure elements, our analysis yielded also interesting information on less ordered traits. Intrinsically flexible elements, i.e. regions characterised by multiple conformers, can be identified unequivocally by their thermodynamic parameters, without recurring to interpretative mechanisms.

Another important point is that we observed a clear difference between parameters corresponding to the cold and the heat denaturation processes: residues that are outliers from the average stability curve tend to have a strong stabilisation effect at low temperature and a weaker destabilising effect at high temperature. This is a strong confirmation that the mechanisms of the two transitions are

intrinsically different according to the mechanism of cold unfolding proposed by Privalov. In this model, cold denaturation is intimately linked to the hydration of hydrophobic residues of the core[1] and with his suggestion that the disruption of the hydrophobic core at low temperature would be caused by the hydration of hydrophobic residue side chains of the core, whereas the high-temperature transition is mainly linked to entropic factors, consistent with the increase of thermal motions when the temperature is increased. This is what we observed in our NMR analysis of Yfh1 and is in line with our previous evidence that showed that the unfolded species at low temperature has a volume higher than the folded species and of the high temperature unfolded species[8] and that cold denaturation is caused by a hydration increase[3].

We also observed, more surprisingly, that some exterior residues, which are certainly well hydrated throughout the whole unfolding process (before, during and after), seem to remain folded at temperatures at which residues of the hydrophobic core appear unfolded. This observation is consistent with the fact that the main mechanism of cold unfolding is the abrupt hydration of hydrophobic residues of the core. This observation, albeit not a definite demonstration of the mechanism of cold unfolding gives a strong indication. This would imply that, at low temperature, the opening of the hydrophobic core and its disruption could happen before the collapse of external and more exposed elements: the core would unfold in lowering the temperature whereas outer turns could be affected last.

## Conclusions

In conclusion, we have provided here an example of a protein that only apparently follows a simple two-state (folded/unfolded) statistical model and for which a global denaturation curve simply hides a profound intrinsic heterogeneity. We described in detail how the unfolding of Yfh1 is a much more complex process than a two-step global unfolding both at high and low temperatures. We are of course aware that, while the global folding/unfolding of Yfh1 can be described by a simple two-state model, NMR shows a more complex picture at the residue level, a situation reported experimentally also in previous publications[14,47]. However, the site-specific analysis of the previous studies remains fundamentally different from ours. The approach suggested by Sborgi et al.[47] is, for instance, of much less general validity, as it refers to a specific case of unfolding: it is one of the few examples of "downhill unfolding" that was discovered and described in the laboratory of Victor Munoz. The paper by Grassein et al.[14], although not yielding sound thermodynamic parameters for single residues, is indeed directly relevant for this paper. Our data clearly show how, as advocated by Grassein et al.[14], local nativeness is probe dependent and, as such, needs to be studied at the individual residue level. The possibility of studying the process relied in our case on the nearly unique properties of Yfh1 but also, more in general, on the use of NMR which is probably the most suitable technique to analyse the contributions to the (un)folding process in a residue-specific manner. It is of course unusual to find a protein, like Yfh1, which shows cold and heat unfolding above water freezing. However, all proteins undergo cold denaturation, albeit at temperatures that do not allow easy detection. We are confident that it is possible to find ways to measure site-specific stability curves for many more instances. We can certainly state that monitoring protein unfolding by the stability curves of individual residues, as allowed by 2D NMR spectroscopy, yielded a much more informative picture than what may have been obtained by any other traditional method. Our work thus paves a new way to the study of protein unfolding that will need to be explored in the future using a number of completely different systems to reconstruct a more complete picture of the complexity of the process.

## Methods

**Sample preparation**. Yfh1 was expressed in BL21(DE3) *E. coli* as previously described[2].

To obtain uniformly $^{15}$N-enriched Yfh1, bacteria were grown in M9 using $^{15}$N-ammonium sulfate as the only source of nitrogen until an OD of 0.6–0.8 was reached and induced for 4 h at 310 K with IPTG. Purification required two precipitation steps with ammonium sulfate and dialysis followed by anion exchange chromatography using a Q-sepharose column with a NaCl gradient. After dialysis, the protein was further purified by chromatography using a Phenyl Sepharose column with a decreasing gradient of ammonium sulfate.

**NMR measurements**. 2D NMR $^{15}$N-HSQC experiments were run on a 700 MHz Bruker AVANCE spectrometer. $^{15}$N-labelled Yfh1 was dissolved in 10 mM HEPES at pH 7.5 to reach 0.1 mM with 0.1 mM selectively $^{15}$N-labelled tyrosine CyaY. Spectra were recorded in the range 278–313 K with intervals of 2.5 K and using the Watergate water suppression sequence[41]. For each increment, eight scans were accumulated, for a total of 240 increments (TD). Spectra were processed with NMRPipe and analysed with CCPNMR software. Gaussian (LB −15 and GB 0.1) and cosine window functions were applied for the direct and indirect dimensions respectively. The data were zero-filled twice in both dimensions. Spectral assignments of Yfh1 were taken from the BMRB deposition entry 19991.

**Selection of the amides to be used in our analysis**. Yfh1 contains 114 backbone amide protons. The first 23 residues are intrinsically disordered[42] and are part of the signal peptide for mitochondrial import, leading to 91 resonances in the globular domain. Sixty-eight residues have nonoverlapping and isolated resonances that allow easily detectable and reliable volume calculation. Most of the excluded overlapping resonances corresponded to disordered regions or to partially unfolded conformations in equilibrium with the folded one in a slow exchange regime at room temperature[7].

**Calculations of the RAD parameters**. The RAD parameter of the backbone amide nitrogen atoms of Yfh1 was calculated on the crystallographic coordinates of a Tyr73-to-Ala mutant solved at 3.0 Å resolution (2fql, Kalberg et al., 2006). This choice was dictated by the better resolution of this structure as compared to an alternative NMR structure (2ga5) or to homology models. The mutation, that is at the very beginning of the globular region of the protein, does not affect the structure of the protein as demonstrated by comparison with other orthologs but changes the self-assembly properties of the protein[43]. No hydrogen atoms were added. RAD was obtained using the software Pops (https://github.com/mathbio-nimr-mrc-ac-uk/POPS) and SADIC (http://www.sbl.unisi.it/prococoa/). The RAD parameter was defined (Puglisi et al., 2020) according to the equation.

$$RAD = (D \times RA \times 100) \tag{2}$$

where D was the distance of an atom from the protein surface as calculated by the programme SADIC[44]. RA was the relative accessibility at atomic level RA defined as the ratio between the exposed surface of a nitrogen atom with respect to that of the whole residue and calculated by the software POP[45]. Most of the exposed residues had RAD values for the amide nitrogens considerably higher than 0.5 and were excluded from the analysis (Table 1). The curves obtained for individual resonances using RAD values between 0.5 and 0.1 had a lower relative spread and a much better agreement with the CD curve (data not shown). The stability curve and the thermodynamics parameters calculated from averaging amide volumes from residues with a RAD value below 0.1 (RAD_0.1) were fully consistent with those calculated from CD spectroscopy, within experimental error[13]. Residues involved in secondary structures were evaluated according to the DSSP programme (https://swift.cmbi.umcn.nl/gv/dssp/)[46].

**Calculation of the stability curves**. Volumes were calculated by summation of the intensities in a set box using the CCPNMR software (https://www.ccpn.ac.uk/v2-software/software). The volumes were normalised by dividing the volume of each peak of Yfh1 at a given temperature by the volume of the CyaY Tyr69 amide peak at the same temperature[13]. This normalisation is meant to filter out the non-linearity of the relationship between peak intensity (or volume) and populations due to instrumental effects. The corrected volumes were transformed into relative populations of folded Yfh1. It is in order to observe that in the quoted paper[13] it was shown that corrections for Yfh1 are only marginal. At each temperature, the fraction of folded protein was estimated by the equation

$$f_U = \left(V_{exp} - V_U\right)/\left(V_F - V_U\right) \tag{3}$$

where $V_{exp}$ is the measured volume, $V_U$ is the volume of the unfolded state (assumed at 313 K), and $V_F$ is the volume of the folded (maximum value) taking into account that, as previously proven[2], at room temperature the unfolded forms of Yfh1 are in equilibrium with the folded population present on average at 70%.

The fraction of folded, $f_F(T)$, and unfolded, $f_U(T)$, forms are a function of the Gibbs free energy of unfolding, $\Delta G°(T)$. If the heat capacity difference between the folded and unfolded forms, $\Delta C_p$, is assumed independent of temperature, the free energy is given by the Gibbs–Helmholtz equation[4]. The thermodynamic parameters $T_m$, $\Delta H_m$ and $\Delta C_p$ were derived by nonlinear least-squares fitting using the Levenberg–Marquardt algorithm from the following equation and omitting the points at 313 K for which, by definition from our assumption, $f_U$ is equal to 1.

$$fU(T) = \frac{e^{-\frac{G°(T)}{RT}}}{1 + e^{-\frac{G°(T)}{RT}}} \tag{4}$$

in which $T_m$, $\Delta H_m$ and $\Delta C_p$ can be obtained by fitting the modified Gibbs–Helmholtz equation

$$\Delta G = \Delta H_m \left[1 - \frac{T}{T_m}\right] + \Delta C_p \left\{(T - T_m) - T \ln\left[\frac{T}{T_m}\right]\right\} \tag{5}$$

The curve corresponding to this equation is known as the stability curve of the protein[18]. Other parameters for low-temperature unfolding, e.g. the low temperature unfolding ($T_c$), were obtained from the stability curve.

Errors on the stability curves were evaluated propagating the errors from the covariance matrix of the fit (Supplementary Figs. S6–S8) and represented as grey lines calculated by the covariance method (Press et al., 1988). They represent how well the measured populations (and thus $\Delta G$) vs. temperature agree with the equation for the stability curve. We reported six representative curves from the subset used to calculate RAD_0.1 (Supplementary Fig. S6), four curves from the subset of Fig. 1c (Supplementary Fig. S7), and four curves corresponding to the best-behaved residues of the beta-sheet (Supplementary Fig. S8). The curves do not fully represent $\Delta G$ because, despite we assumed the protein completely unfolded at 313 K, fitting showed that not all the residues reached a plateau of unfolding at high temperature. We thus indicated the curves as $\widetilde{\Delta G}(T)$ to underline the distinction.

## Data availability

The data that support the findings of this study are available from the corresponding author upon reasonable request.

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

## Acknowledgements

This manuscript is meant in celebration of the 80th birthday of Prof. Robert Kaptein. We thank Geoff Kelly and Tom Frenkiel of the MRC Biomedical NMR Centre for helpful discussions and technical support, Neri Niccolai and Franca Fraternali for help with their software SADIC and PopS, respectively. We are also thankful to Rolf Boelens for his detailed comments and to Jochen Balbach and Dmitry Korzhnev for their constructive criticisms. We acknowledge access to the NMR spectrometers at the Randall unit of King's College London and at the MRC Biomedical NMR Centre in the Francis Crick Institute. The Crick Institute receives its core funding from Cancer Research UK (FC001029), the UK Medical Research Council (FC001029) and the Wellcome Trust (FC001029). The research was supported by UK Dementia Research Institute (RE1 3556) that is funded by the Medical Research Council, Alzheimer's Society and Alzheimer's Research UK.

## Author contributions

All of the authors contributed to the research design and data analyses. R.P. performed most of the experiments described here and of the extraction of the thermodynamic data. G.K. and D.F.H. performed the simulations to estimate the accuracy of the derived stability curves and evaluated the combined effects of the exchange between folded and unfolded, the differential relaxation and the hydrogen exchange. A.P. supported the work financially, provided the protein material and helped writing the manuscript. P.A.T. coordinated the work and wrote the manuscript.

## Competing interests

The authors declare no competing interests.
