## [Peer Review File · Communications Chemistry]

Reviewers' comments:

Reviewer #1 (Remarks to the Author):

This is a very interesting article. It shows application of a basic bio-NMR tool, namely 1H-15N HSQC for evaluation of protein denaturation and folding thermodynamics. The results are important and convincingly presented, and overall the article is well and clearly written. Therefore, without doubt, I recommend publication.

Reviewer #2 (Remarks to the Author):

Puglisi et al. study thermodynamic parameters for cold and heat induced unfolding of the Yfh1 protein using 2D NMR experiments. In a recent communication the same authors have introduced a residue-level analysis of protein folding based on 2D 1H,15N NMR correlations, where amide signals are selected as readouts of cooperative un/folding (RAD approach). Thermodynamic parameters are derived from the signal intensities of folded and unfolded signals, which must be spectroscopically resolved. Corrections for signal intensity changes due to temperature dependent differences of NMR parameters (i.e. linewidth) are considered based on a reference protein. The previous study showed that NMR signals of amides in the core, i.e. remote from the surface, upon correction of temperature effects can reproduce protein stability curves by other methods (assuming 2-state cooperative folding). The present manuscript extends this study by analyzing NMR signals of amides in the exterior region of the folded protein to assess if they could provide additional information about thermodynamic feature of the un/folding process. The authors find that more exterior residues experience additional folding mechanisms compared to those in the interior. Thus analyzing single NMR signals as readout of the overall cooperative folding process (only) is therefore not recommended, as more complex features are expected for surface residues. The authors show that residues in the flexible tails of the protein can be identified and discriminated by their thermodynamic parameters.

Overall the work is technically sound, and provides some interesting information, that goes beyond the previous work by the same authors. Yfh1 seems a very special protein and most proteins will not allow a similar analysis as analysis of NMR spectra at cold and heat induced unfolding temperatures will not be possible. Nevertheless, the potential to extract thermodynamic parameters of cold and heat induced unfolding is certainly relevant and interesting to understand biophysical principles of protein folding and the chosen example provides interesting insight.

Specific comments:

1. The authors "correct" for temperature-dependent effects by comparison with a reference protein, but details of this are not provided. I understand this correction assumes that the reference protein is fully stable in the temperature regime (i.e. does not undergo un/folding? Amino acid composition will still be different and could there not be residue-specific differences that affect the correction?
2. Can the authors rule out the differences seen in stability curves for low temperatures for the exterior residues could reflect a systemic error in the applied correction?
3. The deviations of thermodynamic parameters for residues that are outside the average stability curve at low temperature are discussed to reflect different unfolding mechanisms and be consistent with cold unfolding being driven by hydration of hydrophobic residues. How exactly should this prove the model for cold denaturation, what is the experimental support?
4. The approach shown in the paper can only be used for proteins undergoing cold and warm

denaturation within an NMR accessible temperature range. This is not the case for the vast majority of protein, limiting the general applicability.

5. The datasets and extraction of thermodynamical parameters of unfolding reported in the current manuscript are seemingly the same as previously published (Puglisi et al. 2020), with an additional estimation of experimental error. The authors propose a re-evaluation of their data focusing, this time, on outliers rather than on residues describing well the overall collapses of the protein.

6. With this manuscript, the authors show that despite the unfolding of Yfh1 is well described by a simple two-state model, many residues, mostly included in loops but not only, do not follow this behavior. While this is interesting, this type of behavior, in which the global folding/unfolding can be described by a simple two-state model but for which NMR shows a more complex picture at residue level as been shown experimentally in previous publications (Sborgi et al. 2015, Grassein et al. 2020).

7. The more complex behavior for surface residues and especially tryptophan side chains might simply reflect local events, i.e. local interactions of the aromatic side chains. So, does this really contradict an overall and global 2-state folding process?

8. Figure legends and annotation of figures are sparse and it is not always clear what exactly is shown, also using subpanel labels (a,b,c,...) would be helpful. Examples: Fig S1 (what are the colors?), Fig. S2 (what is shown, error bars?), Fig. S3: what is shown, figure is too small, show a zoomed view), Fig. S4: what is shown in the bottom row?

Reviewer #3 (Remarks to the Author):

The manuscript of Rita Puglisi follows a series of reports on the folding properties of Yfh1, a protein that presents the unique feature of having a cold denaturation temperature above 0°C, enabling its observation using spectroscopic methods. Previously, the authors demonstrated that the intensities measured on 1H-15N HSQC spectra recorded as a function of temperature provide access to the partition between folded and unfolded states of the protein. Choosing appropriate residues as reporters of the protein fold (such as residues located within the hydrophobic core), they demonstrated a very good agreement between the stability curves obtained from Circular Dichroism and NMR measurements. In this manuscript, they take advantage of the residue specific nature of NMR measurement to further their study and examine how the different protein sites report on the melting event. This is performed by clustering residues according different criteria such as their topological properties (defined as the solvent accessibility or the belonging to a given secondary structure element) or the discrepancy of a residue specific stability curves from the global one. The largest discrepancies from global behaviour were found for residues at the protein surface. For these residues, the shape of the stability curve advocates for distinct denaturation mechanism, an observation that is consistent with Privalov's theory. In its current version, this manuscript constitutes a minor incremental step in the study of folding mechanisms of the Yfh1 model protein. The comparison of individual stability curves with the global one previously reported by the authors is a trivial followup of the previous study that suggests some key residues whose contribution may be probed by site specific mutagenesis. For instance, mutagenesis could be used to further investigate the counter intuitive behaviour of the two tryptophan residues. Other points that the authors may consider are listed below.

- The stability curves are derived from the analysis of amide resonances and may therefore contain additional contributions from the exchange with the water protons. Is it possible to conduct the study using non-exchangeable resonances from 1H-13C correlation peaks ? For instance, methyl

resonances from residues at the protein surface or within the hydrophobic core may provide sensitive probes to the protein fold and may allow to assess the contribution of solvent exchange to the fold/unfold partition function.

- The paper is essentially based on the ability to define clusters of residues according the comparison of stability curves derived from their amide resonances. The clustering is performed using either a-priori criteria (such as the RAD parameter or the location on a given secondary structure) or based on the comparison of stability curves of individual residues with the average stability curve. The partition is based on a threshold value of the two melting temperatures (high and low Temp.). The authors provide an interesting theoretical work as supplementary material demonstrating that the only the stability temperature T_s is robustly estimated from the ^1H - ^{15}N correlation. Why then not using this later temperature for the clustering ? Further, a number of clustering methods are currently available that are based on solid statistical framework that may be used to avoid possible bias in the a-priori selection of residues. A distance between two experimental stability curves may be defined and may provide a more solid statistical ground to discuss the findings.

- In figure S1, the correlation at (10.1/134 ppm) displays an unexpected non-linear behaviour that is not discussed. This may indicate an intermediate state in the folding/unfolding path. I was wondering if this correlation belong to one of the two tryptophan residues and if other residues display such non-linear behaviour.

- The list of residues whose stability curves are either consistent or, on the contrary, display significant deviation from the average stability curve shows interesting feature that has not been discussed by the authors, but may suggest mutagenesis strategies to study this system: residues compatible with the global stability curve contain only hydrophobic (as expected) or negatively charged residues (with a single exception of K168) while the other group clusters two histidines (that may be positively charged) in addition to 2 positively charged lysines. This partition raises a number of questions on the folding mechanisms that may be investigated using the strategy developed by the authors combined with an appropriate mutagenesis strategy.

Reviewer #1

We thank the reviewer for these words.

Reviewer #2

[Overall the work is technically sound, and provides some interesting information, that goes beyond the previous work by the same authors. Yfh1 seems a very special protein and most proteins will not allow a similar analysis as analysis of NMR spectra at cold and heat induced unfolding temperatures will not be possible. Nevertheless, the potential to extract thermodynamic parameters of cold and heat induced unfolding is certainly relevant and interesting to understand biophysical principles of protein folding and the chosen example provides interesting insight.]

We thank the reviewer for the global evaluation of our work.

We certainly agree with the reviewer that Yfh1 is a somewhat special protein and, as such, it allowed a series of investigations that could be not normally carried out on more stable proteins. However, theory tells us that all proteins undergo cold denaturation, but most of them do it at temperatures that do not allow detection. This means that Yfh1 is far from being a unique system and it is only reasonable that we extract information from those cases that allow us to observe a phenomenon more clearly. We are actively trying to develop approaches to extend our conclusions to proteins that show only heat induced unfolding above water freezing. We have added a sentence in the discussion to clarify this point.

[1. The authors 1C;correct 1D; for temperature-dependent effects by comparison with a reference protein, but details of this are not provided. I understand this correction assumes that the reference protein is fully stable in the temperature regime (i.e. does not undergo un/folding? Amino acid composition will still be different and could there not be residue-specific differences that affect the correction?)]

We agree with the reviewer: the choice of a single residue from a different reference protein to correct temperature effects could in principle introduce systematic errors. However, we think that they are minimal in our case because the stability of the chosen reference protein (CyaY) in the whole temperature range has been well tested and is clear. The observed effects are thus due only to instrumental sources. We did not mention details of the correction mainly because they were already published in the mentioned paper (Puglisi et al., 2020): we observed in the same paper that corrections for Yfh1 are practically negligible. We have now hopefully clarified this point in the manuscript.

[2. Can the authors rule out the differences seen in stability curves for low temperatures for the exterior residues could reflect a systemic error in the applied correction?]

The reviewer is in principle right. However, as explained in the previous answer, we found that corrections based on the reference protein (CyaY) had a negligible effect on the global unfolding parameters. Therefore, it is very unlikely that they may have a specific influence on exterior residues.

[3. The deviations of thermodynamic parameters for residues that are outside the average stability curve at low temperature are discussed to reflect different unfolding mechanisms and be consistent with cold unfolding being driven by hydration of hydrophobic residues. How exactly should this prove the model for cold denaturation, what is the experimental support?]

We apologise for not been clearer on this point. We observed that some exterior residues, which are certainly well hydrated throughout the whole unfolding process (before, during and after), have stability curves consistent with their “folded” status at temperatures for which residues of the hydrophobic core appear unfolded. This observation is consistent with the fact that the main mechanism of cold unfolding is the abrupt hydration of hydrophobic residues of the core. Of course, this is not, strictly speaking, a definite “proof” but is a strong indication. Altogether, the investigation of residues far from the hydrophobic core might not have a rigorous thermodynamic meaning but it can hint at important tendencies. We have done our best to clarify this point in the text.

[4. The approach shown in the paper can only be used for proteins undergoing cold and warm denaturation within an NMR accessible temperature range. This is not the case for the vast majority of protein, limiting the general applicability.]

The reviewer is right. As explained in the first answer, we are using here an “easy” case to enter in a difficult field. In the future, it may not be impossible to analyse individual residues by NMR methods also for proteins whose cold denaturation is not observable above water freezing. We are actively studying several experimental approaches to extend the applicability of our method.

[5. The datasets and extraction of thermodynamical parameters of unfolding reported in the current manuscript are seemingly the same as previously published (Puglisi et al. 2020), with an additional estimation of experimental error. the authors propose a re-evaluation of their data focusing, this time, on outliers rather than on residues describing well the overall collapses of the protein.]

The reviewer is right, but there is nothing wrong in using the same data for looking at completely different aspects of a phenomenon. The whole structural bioinformatics, for instance, is based mainly on already published datasets (PDB structures). In our case, the previous paper was focused on the possibility of using NMR to obtain unfolding parameters consistent with those obtained with other (commonly used) spectroscopic techniques. The present paper tackles a completely new and ambitious goal: the examination of the unfolding process as observed from external residues. It is an unexplored field because these residues do not take part (in principle) to the two-state unfolding process, yet they “observe” what is going on during the unfolding process. We show that it is possible to measure thermodynamic parameters that, by not representing the global unfolding process in an all-or-none fashion, shed light on the possible mechanisms of unfolding.

[6. With this manuscript, the authors show that despite the unfolding of Yfh1 is well described by a simple two-state model, many residues, mostly included in loops but not only, do not follow this behavior. While this is interesting, this type of behavior, in which the global folding/unfolding can be described by a simple two-state model but for which NMR

shows a more complex picture at residue level as been shown experimentally in previous publications (Sborgi et al. 2015, Grassein et al. 2020).]

The reviewer is partially right. Of course, our work does not come from scratch and we are well aware of previous attempts. However, the interesting paper by Sborgi et al. (2015) is of much less general validity, as it refers to a very specific case of unfolding: it is one of the few examples of “downhill unfolding” that was discovered and described in the laboratory of Victor Munoz. Besides, the residue specific analysis, although accurate, is indirect as the authors did not measure stability curves for individual residues. The paper by Grassein et al. (2020) is indeed very interesting. The quotation of this paper is outstanding in our manuscript. However, also in this case, the approach to single residues is fundamentally different from ours. We have now added explicit reference to the Munoz’s paper.

[7. The more complex behavior for surface residues and especially tryptophan side chains might simply reflect local events, i.e. local interactions of the aromatic side chains. So, does this really contradict an overall and global 2-state folding process?]

The reviewer is right, but we never intended contradicting the existence of an overall global two-state folding process. We wanted instead to examine the behaviour of residues NOT directly involved in the overall unfolding process. The behaviour of some of these residues may reflect only local environment whereas those of other exterior residues seems to shed light on a more complex behaviour.

[8. Figure legends and annotation of figures are sparse and it is not always clear what exactly is shown, also using subpanel labels (a,b,c, 26;) would be helpful. Examples: Fig S1 (what are the colors?), Fig. S2 (what is shown, error bars?), Fig. S3: what is shown, figure is too small, show a zoomed view), Fig. S4: what is shown in the bottom row?]

The reviewer is right. We did our best to improve the figures in the revised manuscript. In particular, we made subpanel labels more visible and cited the subpanels explicitly throughout the manuscript. We changed Figure S3 completely, making the chosen peaks more visible in their zoom. As for figures S1 and S2, they are meant as examples to clarify the text. It would only burden the file to add many more details. We have nevertheless clarified the legends. The same for Figure S4.

Reviewer #3

[The comparison of individual stability curves with the global one previously reported by the authors is a trivial followup of the previous study that suggests some key residues whose contribution may be probed by site specific mutagenesis. For instance, mutagenesis could be used to further investigate the counter intuitive behaviour of the two tryptophan residues. Other points that the authors may consider are listed below.]

We respectfully disagree with this comment. The present investigation of individual stability curves is indeed a follow-up of our previous NMR study, as clearly indicated in our introduction. However, it is by no means trivial. It is the first experimental attempt to measure stability curves of individual residues of a protein, including many exterior residues not directly involved in the 2-state unfolding process. As for the mutagenesis

suggestion, it is of course a good suggestion in principle, but it may be difficult to apply it. In the past, we carried out several successful mutagenesis investigations on this protein, proving among other things a possible molecular origin of cold denaturation (see Sanfelice et al., *Chemphyschem*. 2015 Dec 1;16(17):3599-602). However, if applied to the present analysis, a mutagenesis approach would be rather difficult to interpret because any mutation could affect BOTH global and site-specific parameters, possibly in a different way. As for mutation of the tryptophan: the tryptophan in question is indeed exposed but interacts with a spatially close conserved arginine (Arg141) which forms pi-pi interactions with the tryptophan. We have realised in the past that mutation to a non-aromatic residue causes aggregation and unfolding.

We will happy to examine this suggestion in future work to see if it is feasible at all.

[The stability curves are derived from the analysis of amide resonances and may therefore contain additional contributions from the exchange with the water protons. Is it possible to conduct the study using non-exchangeable resonances from 1H-13C correlation peaks ? For instance, methyl resonances from residues at the protein surface or within the hydrophobic core may provide sensitive probes to the protein fold and may allow to assess the contribution of solvent exchange to the fold/unfold partition function.]

The reviewer is in principle right and indeed we did examine additional contributions thoroughly (see in particular Supplementary Information). The suggestion of using non-exchangeable resonances, for instance 1H-13C correlation peaks, is excellent and we will pursue it in the future. However, this approach would report, as the reviewer him/herself says, on unfolding from a different angle. The value of looking at 15N HSQCs is precisely the possibility to observe solvent exchange protons as the solvent seems to be an important component of cold denaturation.

[The paper is essentially based on the ability to define clusters of residues according the comparison of stability curves derived from their amide resonances. The clustering is performed using either a-priori criteria (such as the RAD parameter or the location on a given secondary structure) or based on the comparison of stability curves of individual residues with the average stability curve. The partition is based on a threshold value of the two melting temperatures (high and low Temp.). The authors provide an interesting theoretical work as supplementary material demonstrating that the only the stability temperature T_s is robustly estimated from the 1H-15N correlation. Why then not using this later temperature for the clustering? Further, a number of clustering methods are currently available that are based on solid statistical framework that may be used to avoid possible bias in the a-priori selection of residues. A distance between two experimental stability curves may be defined and may provide a more solid statistical ground to discuss the findings.]

The reviewer is in principle right. It would, in principle, be possible to use T_s instead of the two unfolding temperatures (T_c and T_m). The reason why we preferred to use the two temperatures is that T_s reports only on a specific temperature, whereas we report on the whole area defined by the temperature curve As explained in the main text, according to the rough classification of Nojima et al. (1977), altered thermostability can be achieved thermodynamically by three extreme mechanisms (Figure 3). T_s is influenced mainly by only one of these mechanisms whereas T_m and T_c depend essentially from all three mechanisms. In other words, considering both T_m and T_c implies taking into account the whole stability curve. This is an important point as we

have demonstrated in the past (Alfano et al., *Nat. Commun.*, 8, 15428 (2017).) that the best parameter to describe protein stability is the whole area of the stability curve. We are in principle in favour of other clustering methods, but the choice is arbitrary. We hope instead that our work will encourage other researchers to investigate this aspect. We shall take great care of the suggestion for future work on the subject presently actively undergoing in our lab.

[In figure S1, the correlation at (10.1/134 ppm) displays an unexpected non-linear behaviour that is not discussed. This may indicate an intermediate state in the folding/unfolding path. I was wondering if this correlation belong to one of the two tryptophan residues and if other residues display such non-linear behaviour.]

This is a valid observation. Non linear trends of CS as a function of temperature are undoubtedly interesting. However, the non linearity in the peak of Met109, as in some other residues, is very minor. We could speculate on its meaning but this is how data have been analysed over the last 30-40 years. Our challenge here is to propose a completely different way to analyse unfolding using a somewhat “special” case of a protein that allows determination of the whole curve of stability. We could use the old approach but what would be then the advantage to be able to work on something that opens up a whole new lot of possibilities?

[The list of residues whose stability curves are either consistent or, on the contrary, display significant deviation from the average stability curve shows interesting feature that has not been discussed by the authors, but may suggest mutagenesis strategies to study this system: residues compatible with the global stability curve contain only hydrophobic (as expected) or negatively charged residues (with a single exception of K168) while the other group clusters two histidines (that may be positively charged) in addition to 2 positively charged lysines. This partition raises a number of questions on the folding mechanisms that may be investigated using the strategy developed by the authors combined with an appropriate mutagenesis strategy.]

As discussed above in answering to the general observations of this reviewer, the suggestion of using mutagenesis is certainly interesting. At present, we chose not to follow it, not only to avoid an extra lengthening of this already long manuscript but also because, in principle, it might be difficult to unravel the relationship between global and local parameters for any mutation; it would certainly be a long, albeit interesting, work. Each mutation would lead to an independent investigation by itself. We plan to tackle the feasibility of this approach in the near future.

REVIEWERS' COMMENTS:

Reviewer #2 (Remarks to the Author):

The authors have responded to all the points raised and addressed them by some text changes. Figures legends could still be a bit more clear but are improved compared to the previous version. I recommend publication.

REVIEWERS' COMMENTS:

Reviewer #2 (Remarks to the Author):

The authors have responded to all the points raised and addressed them by some text changes. Figures legends could still be a bit more clear but are improved compared to the previous version. I recommend publication.

We thank the reviewer.

We did try to further improve Figure legends.